

# Kinematic parameters of internal waves of the second mode in the South China Sea

Oxana Kurkina[1], Tatyana Talipova[1,2], Tarmo Soomere[3,4], Ayrat Giniyatullin[1], Andrey Kurkin[1]

[1]Nizhny Novgorod State Technical University n.a. R.E. Alekseev, Nizhny Novgorod, 603950 Russia
[2]Institute of Applied Physics, Nizhny Novgorod, 603950 Russia
[3]Department of Cybernetics, School of Science, Tallinn University of Technology, Tallinn, 12618 Estonia
[4]Estonian Academy of Sciences, Kohtu 6, Tallinn, 10130 Estonia

*Correspondence to*: Andrey Kurkin (aakurkin@gmail.com)

**Abstract.** Spatial distributions of the main properties of the mode function and kinematic and nonlinear parameters of internal waves of the second mode are derived for the South China Sea for typical summer conditions in July. The calculations are based on the Generalized Digital Environmental Model (GDEM) climatology of hydrological variables. The focus is on the phase speed of long internal waves and the coefficients at the dispersive, quadratic and cubic terms of the weakly nonlinear Gardner model. Spatial distributions of these parameters, except for the coefficient at the cubic term, are

qualitatively similar for waves of both modes. The dispersive term of Gardner equation and phase speed for internal waves of the second mode are about a quarter and half, respectively, of those for waves of the first mode. Similarly to the waves of the first mode, the coefficients at the quadratic and cubic terms of Gardner equation are practically independent of water depth. In contrast to the waves of the first mode, for waves of the second mode the quadratic term is mostly negative. The results can serve as a basis for express estimates of the expected parameters of internal waves for the South China Sea.

Keywords: Internal waves, Long waves, Solitary waves, Gardner equation, South China Sea

## 1 Introduction

The South China Sea is an example of shelf seas where highly energetic internal solitary waves often generate up to 100–200 m vertical displacements of water masses. These powerful disturbances are usually excited by interactions of barotropic

tidal waves with the Kuroshio Current and further modified by numerous islands, seamounts and other bathymetric features in the Luzon Strait (Liu et al., 1998, 2004, 2006; Cai et al., 2002; Rump et al., 2004, 2015). Many such structures with amplitudes up to 100 m resemble solitons (solitary waves that interact elastically). Their impact on water masses has been observed in the vicinity of two underwater elevations in an area of the Luzon Strait where the water depth is only about 300 m (Liu et al. 2006). These waves propagate into deeper regions of the South China Sea, cross this water body and exert

conspicuous transformations along the continental shelf at depths of 400–200 m. The associated displacements of isopycnals

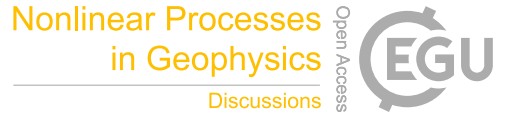

may reach 100 m. The appearance of such waves often matches well theoretical shapes of internal solitons (Klymak et al., 2006).

While most ~~of~~ such waves represent internal waves of the first mode, numerous recordings suggest that internal waves of the second and sometimes even third mode are regularly present in the South China Sea (Fig. 1; see also Guo et al., 2006; Yang

et al., 2009; Vlasenko et al., 2010; Liu et al., 2013). Higher modes of long internal waves are apparently relatively often generated in the World Ocean. They are frequently excited in areas where large-scale flows interact with sizeable underwater bathymetric features with steep gradients (Ramp et al., 2010, 2015; Shroyer et al., 2010; Vlasenko et al., 2010, 2014; Vlasenko and Stashchuk, 2015). Alternatively, internal waves of any mode may be created in micro-tidal stratified semi-sheltered basins by an intense outflow or inflow (Vlasenko et al., 2009), strong atmospheric disturbances (Ivanov et al.,

1987), release of storm surges and other phenomena.

Most of studies of internal waves focus on waves of the first mode. This approach apparently mirrors the abundance of records of various properties of water masses in the uppermost layers of the ocean compared to profiles of the entire water column. Namely, profiles of motions and other hydrophysical properties of the upper sections of water masses usually do not provide enough information about the full vertical structure of internal waves. Separation of the internal wave field into

components representing different modes is extremely complicated in the vicinity of their generation regions and in areas where the waves interact with one another and with the bottom. Such analysis, however, is feasible in regions remote from the generation and interaction areas because small-amplitude internal waves of different modes propagate with different velocities and become separated after some time.

Solitary internal waves of the first mode may be waves of elevation or waves of depression. The structure of higher-mode

internal solitary waves is more complicated (Fig. 2). For example, vertical displacements of the upper and lower jump layers created by an internal solitary wave of the second mode have different polarities. For this reason the notion of internal waves is based on certain topologic features of the instantaneous appearance of the intermediate layer. Waves that create convex modifications of this layer are said to have positive polarity. Such waves are called convex or positive waves in what follows. Waves that create a concave shape of the intermediate layer are said to have negative polarity and are called

concave or negative waves.

Internal solitary waves of the second mode with both polarities have been regularly observed on the north-western continental shelf of the South China Sea (Yang et al., 2010; Ramp et al., 2015). Such waves may be generated during interactions of solitary waves of the first mode with various bathymetric features (Vlasenko and Hutter, 2001). Positive (convex) solitary waves of this kind appear substantially (by about 20 times) more frequently in the records than negative

(concave) waves (Yang et al., 2010).

The dynamics of long internal waves of the second mode can be described with reasonable accuracy using weakly nonlinear evolution equations of the Korteweg–de Vries (KdV) family. In particular, Gardner equation is commonly used as the classic model of internal waves of the first mode (Holloway et al., 1999; Grimshaw et al., 2004; Talipova and Pelinovsky, 2013; Talipova et al., 2014, 2015). This model has been applied *inter alia* to explain and replicate a polarity switch of internal



solitons propagating along the north-eastern continental shelf in the South China Sea (Liu et al., 1998; Orr and Mingerey, 2003; Zhao et al., 2004; Grimshaw et al., 2010). The models of this kind are correct only asymptotically. Their core advantage is that a small set of parameters governs the appearance and properties of internal solitary waves. This feature makes it possible to use these models to isolate and identify principally new features of the dynamics of internal waves even

if some details of the system are not reproduced (Kurkina et al., 2016; Lamb et al., 2007).

The parameters selected for the model have a significant effect on many features of internal solitary waves. In other words, the appearance and core qualities of the propagation and transformation of such waves are governed by spatial variations in the coefficients of Gardner equation along the propagation path of the waves in question. The associated variations have been thoroughly studied for internal solitary waves of the first mode using common data bases of the vertical structure of

temperature and salinity (Levitus, 1982; Garnes, 2009). This approach made it possible to construct climatologically valid maps of spatio-temporal variations in various coefficients of Gardner equation for internal waves of the first mode in different regions of the World Ocean. These maps depict the values of phase speed of long waves (also called wave speed because for long waves it is also equal to group speed) and coefficients of various terms (linear, quadratic and cubic term) in the relevant Gardner equation (Pelinovsky et al., 1995; Talipova et al., 1998; Talipova and Polukhin, 2001; Polukhin et al.,

2003, 2004; Kurkina et al., 2011, 2017). Similar maps have been also calculated for the South China Sea (Grimshaw et al., 2010; Liao et al., 2014).

As many regions of the World Ocean support propagation of highly energetic internal waves of higher modes, it is important to expand this kind of 'climatology' of internal wave propagation regimes to cover, to a first approximation, the properties of large-amplitude internal waves of the second mode. Such maps of the kinematic parameters (wave speed and the coefficient

at the linear term) and coefficients at the nonlinear terms of the relevant evolution equation make it possible to rapidly evaluate several core properties of the dynamics and impact of internal waves, build pathways of the propagation of waves from their typical areas of generation and identify which regions are possibly affected by hydrodynamic loads created by large internal waves.

For example, the polarity of solitons is governed by the sign of the coefficient at the quadratic term of Gardner equation

(Grimshaw et al., 2007). The values of this coefficient as well as other kinematic parameters of waves can be calculated in a straightforward manner from the so-called mode function and its derivatives. The lines where the coefficient at the quadratic term vanishes or changes its sign mark the regions of a switch of the polarity of internal solitons. This switch may be accompanied by radical changes in the further behaviour of waves or the region may even be a location of the onset of wave breaking. This feature is valid for solitons of the first and second modes. Similar maps of the values of the coefficient at the

cubic term specify, e.g., the regions where modulational instability of internal wave trains may modify wave properties or where a specific type of solitons – internal breather – may exist (Talipova et al, 2011).

This paper focuses on the construction of maps of phase speed and coefficients at various terms of Gardner equation. These quantities are often called kinematic and nonlinear parameters of long internal waves of the second mode. The target area is the South China Sea where such maps are urgently needed to better evaluate the core properties of internal waves and their





propagation. We start with a short description of the setup of the problem of internal wave propagation. An asymptotic solution to this problem can be provided by an evolution equation for such internal waves – Gardner equation. To properly evaluate the values of its coefficients that govern the appearance and dynamics of internal waves of the second mode, it is necessary to adequately describe the structure of the mode function. A relevant nonlinear correction to this function is derived using an asymptotic procedure, which is discussed in Section 2 together with the main features of the appearance of internal solitary waves of the second mode. Section 3 describes the resulting maps of phase speed and various coefficients at the nonlinear terms of the Gardner model for internal waves of the second mode and the applicability of the entire model for the conditions of the continental shelf of the South China Sea. The main conclusions of the study are formulated in Section 4.

## 2 Vertical structure of long internal waves of the second mode

Similarly to the treatment of internal waves of the first mode, the dynamics of long internal waves of the second mode in the ocean can be adequately described using a classic evolution equation – Gardner equation (Holloway et al, 1999; Grimshaw et al, 2004, 2007). This model equation, presented here in the nondimensional form:

$$\frac{\partial \eta}{\partial t} + \left(c + \alpha \eta + \alpha_1 \eta^2\right)\frac{\partial \eta}{\partial x} + \beta \frac{\partial^3 \eta}{\partial x^3} = 0 , \tag{1}$$

is valid for internal waves of any mode. Here $x$ denotes distance along the propagation direction of the wave, $t$ is time, $\eta$ is the vertical deviation of the isopycnals from their equilibrium position at a selected vertical location $z^*$, $c$ is the phase speed of internal waves, $\alpha$ and $\alpha_1$ are the coefficients at the quadratic and cubic nonlinear terms, respectively (sometimes also called the quadratic and cubic nonlinear parameters), and $\beta$ is the coefficient at the linear dispersive term (often called the dispersion coefficient). The quantities $c$, $\alpha$, $\alpha_1$, and $\beta$ represent the major kinematic characteristics of the internal wave field.

The mode function $\Phi$ is an eigenfunction of the Sturm–Liouville problem:

$$\frac{d^2 \Phi}{dz^2} + \frac{N^2(z)}{c^2}\Phi = 0, \tag{2}$$

where $N(z)$ is the buoyancy (Väisälä–Brent) frequency. This frequency depends on the local stratification and is defined as follows:

$$N^2(z) = -\frac{g}{\rho(z)}\frac{d\rho(z)}{dz} , \tag{3}$$

where $g$ is acceleration due to gravity, $\rho(z)$ is the undisturbed density profile, and $H$ is the total water depth. The boundary conditions for Eq. (3) usually include the requirement of the vanishing of $\Phi$ at the bottom and at the sea surface. We chose the common approximation of the so-called rigid lid at the surface, for which these conditions reduce to $\Phi(0) = \Phi(H) = 0$.





The phase speed $c$ of long internal waves is an eigenvalue of the described Sturm–Liouville problem. The vertical location $z^*$ is, theoretically, arbitrary but the resulting numerical values of the coefficients of Gardner equation (1) obviously depend on the choice of $z^*$.

We follow the tradition to select $z^*$ at the location $z_{max}$, which corresponds to the maximum value $\Phi_{max}$ of the mode function $\Phi$ (Holloway et al., 1999). This function is normalised as $\Phi(z^*) = 1$. Details of the spatio-temporal structure of internal waves are described in this model as

$$\zeta(x,z,t) = \eta(x,t)\Phi(z) + \eta^2 F(z).\tag{4}$$

Here $F(z)$ has the meaning of a second-order nonlinear correction to the mode function. It is defined as a solution of the following inhomogeneous boundary problem:

$$\frac{d^2F}{dz^2} + \frac{N^2}{c^2}F = -\frac{\alpha}{c}\frac{d^2\Phi}{dz^2} + \frac{3}{2}\frac{d}{dz}\left[\left(\frac{d\Phi}{dz}\right)^2\right], \quad F(0) = F(H) = 0.\tag{5}$$

A unique solution for Eq. (5) can be obtained using an additional normalising condition $F(z^*) = 0$. Even though the location $z^*$ can be chosen arbitrarily, the resulting vertical structure of motions and displacements of different water parcels are invariant with respect to the particular choice of $z^*$ (Holloway et al., 1999, 2001). The coefficients of Eq. (1) can be expressed explicitly for any stratification in terms of the mode function, its derivatives and integrals:

$$\beta = \frac{c}{2D}\int_0^H \Phi^2 dz, \qquad \alpha = \frac{3c}{2D}\int_0^H \left(\frac{d\Phi}{dz}\right)^3 dz, \qquad D = \int_0^H \left(\frac{d\Phi}{dz}\right)^2 dz,\tag{6}$$

$$\alpha_1 = \frac{1}{2D}\int_0^H dz\left\{9c\frac{dF}{dz}\left(\frac{d\Phi}{dz}\right)^2 - 6c\left(\frac{d\Phi}{dz}\right)^4 + 5\alpha\left(\frac{d\Phi}{dz}\right)^3 - 4\alpha\frac{dF}{dz}\frac{d\Phi}{dz} - \frac{\alpha^2}{c}\left(\frac{d\Phi}{dz}\right)^2\right\}.\tag{7}$$

Consequently, for the specifying of the coefficients of Gardner equation it is necessary to evaluate the mode function from Eq. (2) and its nonlinear correction from Eq. (5).

The vertical structure of internal waves of the second mode is more complicated than the same structure for the classic internal waves of the first mode. The core difference between these structures can be illustrated on the example of a simple model of quasi-two-layer stratification (Fig. 3a). The water masses described by such a model have one jump layer of density. The Väisälä frequency (Fig. 3b) has one maximum along each vertical cross-section. The maximum is located in the region of the fastest variation in density. Similarly, the mode function for the waves of the first mode has one extremum (maximum or minimum depending on the normalisation) along each vertical cross-section. Importantly, the mode function for the waves of the first mode exhibits no sign change within the entire water column.

In contrast, the mode function for the waves of the second mode changes its sign. It has the maximum positive value near the upper boundary of the jump layer and the minimum (negative) value near the lower boundary of the jump layer (Fig. 4a).





This means that two more intrinsic quantities are present in the system: the locations of zero-crossing $z_0$ and minimum $z_{min}$ of the mode function. If some function $\Phi$ satisfies Eq. (2) with $\Phi(0) = \Phi(H) = 0$, the function $-\Phi$ is also a valid solution to this boundary problem. Therefore, it is not clear beforehand whether $z_{max}$ or $z_{min}$ should be chosen to normalise the mode function and to specify the unique mode function from the family described by Eq. (2) and the relevant boundary conditions.

There are different approaches in the literature. Liu and Wang (2012) rely on the values of the mode function at its minimum $z_{min}$, where the absolute value of $\Phi$ is usually the largest. This approach, in essence, follows the logic of addressing the dynamics of internal waves of the first mode where the absolute maximum of the mode function is chosen as the scale for the normalisation of this function.

Other recent studies of internal solitons of the second mode in the South China Sea address the situation on the continental

shelf where the largest absolute values of $\Phi$ are located at $z_{max}$ relatively close to the sea surface (Yang et al, 2009, 2010). In such situations it is natural to choose the maximum of $\Phi$ in a location above the jump layer (main pycnocline) as the basis for normalisation. The quality and resolution of measurements above the main pycnocline are often better than in deeper layers and thus the vertical structure of the mode function is more reliably represented.

In such environments it is convenient to adjust Gardner equation so that it describes the deviations of the isopycnals that

correspond to $z^* = z_{max}$. Similarly to the analysis of internal waves of the first mode, the entire mode function $\Phi$ is then normalised so that the global maximum of its absolute values is 1. Also, in this case the zero-crossing point of the nonlinear correction $F$ (Eq. (5)) is linked to $z_{max}$. In this framework the function $\eta$ expresses deviations of the isopycnal $\zeta(x,z,t)$ in Eq. (4) at $z = z_{max}$ from its undisturbed location.

Importantly, with this choice of normalisation the polarity of the internal solitons of the second mode matches the sign of the

20 coefficient at the quadratic term of Eq. (2). In other words, positive (convex) solitons correspond to positive values of the coefficient $\alpha$. This match follows the usual interpretation of the coefficients and the appearance of solutions of the family of generalisations of the KdV equation. This choice, however, may be problematic in the analysis of wave motions of deeper parts of the World Ocean and the South China Sea. The problem becomes evident when both the seasonal and the main pycnocline are located relatively close to the surface. In such cases the absolute value $|\Phi(z_{min})|$ may by several times exceed

the maximum of the mode function $\Phi(z_{max})$. Consequently, the largest negative values of the normalised mode function may reach values well beyond $-1$.

This feature may lead to certain problems in the analysis of the dynamics of internal waves of this kind. However, the normalisation is, in essence, arbitrary and the vertical structure of motions is independent of the chosen normalisation. Therefore, the relevant issues are purely technical and do not impact on the results of the analysis. Thus, we decided to meet

the possible technical implications but still follow the more logical and straightforward normalisation using the maximum of the absolute values of the mode function at the upper boundary of the jump layer.



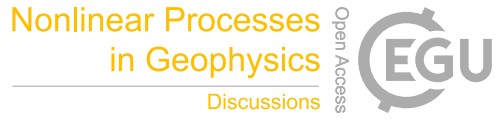

## 3 Kinematic parameters of long internal waves of the second mode

The South China Sea is a large (surface area about 3.5 million km$^2$) semi-sheltered water body bordered by China from the north, Vietnam and Cambodia from the west and the Philippines from the south-east (Fig. 5a). The water depth of this sea varies greatly (Fig. 5b). About half of the sea is located on the continental shelf and has water depths just a few 100's of

5 metres while another half has depths comparable with those of the open ocean. This variability together with extensive variations in the hydrological fields in this sea gives rise to large spatial variations in the coefficients of the underlying nonlinear evolution equations and, consequently in the kinematic parameters of internal waves in the area.

To construct spatial maps of these parameters, we followed the approach implemented for the calculation of similar parameters of internal waves of the first mode and the coefficients of Gardner equation for this basin (Grimshaw et al., 2010;

Liao et al., 2014). We employed generalised climatologic information about long-term mean temperature and salinity profiles. This information was extracted from the Generalized Digital Environment Model (GDEM) (Teague et al., 1990; Carnes et al., 2009). The GDEM provides coefficients of mathematical expressions describing the vertical profiles of temperature and salinity with a horizontal resolution ranging from 30′ in the deep ocean to 10′ in selected coastal regions (incl. the South China Sea) at 77 vertical levels. The relevant database integrates over three million observations since 1975.

The maps in this paper were calculated for the stratification that is characteristic in July. Mean density profiles were computed for each horizontal pixel of the GDEM database from temperature and salinity profiles using the International Equation for State of Sea Water (Fofonoff and Millard, 1983). With these density profiles, we evaluated the mode function $\Phi(z)$ for the second mode as the normalised eigenfunction of the boundary problem (2) similarly to the procedure employed in (Kurkina et al., 2017). The field of large-scale currents was ignored. The eigenvalue problem (2) was solved numerically

at each pixel of the GDEM for the first and the second eigenvalue $c$ (phase speed of long linear internal waves of the first and second vertical mode) and for the first and second eigenfunction $\Phi(z)$ (vertical structure of the wave). Further, the boundary-value problem (5) for the nonlinear correction $F(z)$ was solved numerically by the method of variation of parameters. Finally, the coefficients at the linear, quadratic and cubic nonlinear terms were evaluated from Eqs. (6,7).

### 3.1 Spatial variations in the parameters of the mode function

As described above, the calculations of various parameters and coefficients rely on the values of maxima $z_{max}$, minima $z_{min}$ and zero-crossings $z_0$ of the mode function (Fig. 4a). Figures 6–8 present spatial distributions of these quantities (normalised against the total depth of the sea) and frequency histograms of the occurrence of their different values.

A large part of the values of $z_{max}/H$ are concentrated around $z_{max}/H = 0.97$ (Fig. 6b). This relatively wide peak in the histogram indicates that in a substantial part of the sea the location of the maximum of the mode function is found relatively

close to the sea surface. The pixels with such values evidently belong to the deep-water region of the study area. Another, minor and narrow peak is located at $z_{max}/H = 0.75$. This peak apparently reflects a more or less horizontal sub-region of the sea on the continental shelf. The histogram of $z_0/H$ (Fig. 7b) has a similar shape. The main and secondary peaks in this



histogram are concentrated around slightly smaller values $z_0/H = 0.9$ and $z_0/H = 0.5$, respectively. In contrast, the distribution of $z_{min}/H$ (Fig. 8b) is much flatter but still contains distinct peaks at $z_{min}/H = 0.25$, 0.45 and 0.65, and a narrow peak at $z_{min}/H = 0.2$. These features suggest that interrelations between the vertical locations of the maximum, zero-crossing and minimum of the mode function are greatly different.

The locations of zero-crossings of the mode function largely follow the relevant locations of the maximum. Therefore, both zero-crossings and maxima of the mode function roughly reflect the core variations in the water depth. In contrast, the minima of the mode~~al~~ function are only weakly, if at all, correlated with $z_{max}$ and $z_0$. It is therefore likely that the quantity $z_{min}$ reflects some other features of the bathymetry and hydrography of the sea. This conjecture once more supports the choice of the maximum of the mode function for the normalisation of this function.

Another view of the nature of the distributions of the quantities $z_{max}/H$, $z_0/H$ and $z_{min}/H$ can be provided using a scatter-plot of their values against the physical water depth (Fig. 9). The plots of all three quantities exhibit a cluster with extensive variation in their values for relatively small depths. This feature indicates that very large variability in ~~of~~ kinematic properties of internal waves of the second mode is an intrinsic feature of relatively shallow regions of the study area. For depths larger than 500 m all three quantities show a clear, almost rigorous dependence on water depth. Consistently with the

above, $z_{max}/H$ and $z_0/H$ are concentrated in a narrow range close to 1. Interestingly, $z_{min}/H$ exhibits an almost linear relationship with water depth. This feature signals that the structure of the mode function of the second mode may have a certain systematic pattern of changes along the propagation of internal waves from their generation area over the deep-water region of the South China Sea towards the continental shelf. This pattern does not become visible in the behaviour of the quantities $z_{max}/H$ and $z_0/H$.

As discussed above, it is debatable whether the maximum or minimum values of the mode function should be used for normalising this function. The spatial distribution of the values of $\Phi_{min}/\Phi_{max}$ indicates that the maximum of $\Phi$ exceeds the absolute value of the relevant minimum in most of the relatively shallow-water part of the study area whereas in the deeper regions $|\Phi_{min}|$ is systematically larger than $\Phi_{max}$ (Fig. 10). A histogram of the normalised values of $\Phi_{min}$ contains two peaks of comparable height and width. The peak at $\Phi_{min}/\Phi_{max} = -1$ evidently reflects the typical values of this ratio in the shallow

areas whereas another peak at –2 is characteristic of this ratio in deeper regions. The values of $|\Phi_{min}|$ do not exceed 2.5 in the interior of the South China Sea but reach levels >3 in the Sulu Sea.

### 3.2 Distributions of kinematic parameters of internal waves of the second mode

Spatial distributions of phase speeds of long linear internal waves of the first and second modes are very similar to each other in the South China Sea (Fig. 11). The phase speeds for internal waves of the second mode are mostly below 1.5 m/s in

this water body in typical conditions of July. Internal waves of the first mode propagate much faster. The phase speeds of waves from the first and second modes usually differ by a factor of 1.5–2. As expected, the phase speed of waves of both modes largely depends on the water depth (Talipova and Polukhin, 2001; Polukhin et al., 2003). The presence of two





subregions of the study area with different characteristic phase speeds appears in the scatter-plot of phase speeds and water depth (Fig. 12a). This feature becomes distinctly evident as a two-peak distribution of the empirical distribution of different phase speeds (Fig. 12b). Most internal waves of the second mode propagate with speeds around 0.2 m/s or around 1.4 m/s in the South China Sea.

Even though water depth is one of the most important factors governing the propagation speed of internal waves, stratification of water masses equally contributes to the properties of the propagation of internal waves. The vertical profile of water density to a large extent expresses a balance between the impact of radiation of the Sun with processes that govern the formation of the salinity field in the ocean (e.g., temperature increase in the upper layer and associated evaporation from the surface followed by downward convection of saltier water). These processes heavily depend on the level of incoming
radiation. The changes in the amount of radiation from the Sun may be one of the reasons of the presence of the meridional pattern of the phase speed of internal waves of the second mode. This meridional pattern is well known for internal waves of the first mode (Talipova and Polukhin, 2001). Its presence is a likely reason why the dependence of the phase speed on water depth shows substantial scatter in the study area (Fig. 12a). The level of scatter is, however, fairly moderate and the relationship between the water depth and phase speed can be reasonably approximated using a power function

$$c = q_c H^a \tag{8}$$

For water depths less than 500 m an appropriate approximation of the coefficient in Eq. (8) is $q_c = 0.0078 \text{ m}^{0.25}\text{s}^{-1}$ (Fig. 12a). The 95% confidence interval of this estimate is [0.0075, 0.0081]. The estimate for the power in Eq. (8) is $a = 0.75$ whereas the relevant 95% confidence interval is [0.743, 0.757]. For water depths exceeding 500 m (Fig 12a) respective estimates are $q_c = 0.1901 \text{ m}^{0.756}\text{s}^{-1}$ (95% confidence interval [0.1802, 0.2]) and $a = 0.244$ (95% confidence interval [0.237, 0.251]).

Spatial distributions of the coefficient at the dispersive term of Gardner equation (1) for internal waves of the second (Fig. 13a) and first (Fig. 13b) modes are also qualitatively similar. However, the numerical values of this coefficient differ substantially. This coefficient (and, consequently, the impact of linear dispersion on the wave propagation and dynamics) for waves of the second mode is by about 3–4 times smaller than the similar coefficient for the waves of the first mode. The relationship between this coefficient and water depth (Fig. 13b) is remarkably different from a similar relationship (8) for
phase speed. Figure 13b clearly represents a quadratic relationship that graphically can be presented as a parabola

$$\beta = q_\beta H^2 . \tag{9}$$

An estimate for the coefficient in Eq. (11) is $q_\beta = 0.01682 \text{ m s}^{-1}$, with a 95% confidence interval [0.01669, 0.01695]. The scatter of the values of this coefficient for a given depth increases with the increase in the water depth. This feature demonstrates that deep-water stratification may have a considerable impact on the values of the coefficient at the dispersive
term in deeper areas.

Differently from the coefficient at the dispersive term, the values of coefficients at the nonlinear terms of Gardner equation are mostly governed by properties of stratification and only insignificantly depend on the water depth (Talipova and Polukhin, 2001). It is therefore not surprising that the maps of these coefficients for waves of the second (Fig. 14a) and first



(Fig. 14b) modes are qualitatively similar to each other and that the numerical values of these coefficients for the two modes are comparable.

The histogram of the values of the coefficient at the quadratic term in Eq. (1) indicates that, differently from several other quantities addressed above, this coefficient has a clearly skewed but unimodal distribution. The values with both signs are more or less equally represented (Fig. 14c). The range of values is from $-0.01$ to $+0.02$ s$^{-1}$. The most frequent values are negative, the relevant peak is located at $-0.007$ s$^{-1}$, and the majority of single values are also negative. This feature apparently mirrors the larger extent of deep-water regions compared to relatively shallow ones in the South China Sea. However, the area of the shallow-water region of the sea is also significant and almost half of the values of the coefficient at the quadratic term are positive.

Interestingly, a smaller peak exists for zero values of this coefficient. Gardner equation is not applicable in locations where the coefficient at the quadratic term vanishes and one has to employ a modified KdV equation in order to properly describe weakly nonlinear dynamics of internal waves in such regions. Importantly, the study area contains regions characterised by large gradients and changes in the sign of this coefficient. In general, the signs of this coefficient are different in deeper-water and shallower regions of the South China Sea. Interestingly, the signs of this coefficient are also different in the north-western and south-western regions of the continental shelf for both modes.

The coefficient in question is positive in most of the northern part of the shelf; consequently, the situation is favourable for the existence of positive internal solitons of the second mode. This feature explains why convex solitons are predominantly recorded in the north-eastern segments of the continental shelf (Yang et al., 2010). In contrast, this coefficient is generally negative for internal waves of the first mode whereas its positive values are found only in a few small areas of the South China Sea. Further, this coefficient for waves of the second mode is predominantly positive in the southern relatively shallow part of the South China Sea whereas for waves of the first mode the sign of this coefficient is highly variable.

It is well known that the values of the coefficient of the quadratic term of Eq. (2) for internal waves of the first mode are practically independent of the water depth (Talipova and Polukhin, 2001). This property is also true for internal waves of the second mode in the South China Sea (Fig. 15). The largest absolute values of this coefficient (corresponding to both negative and positive values) occur in relatively shallow areas. The range of its values in deeper parts (depths >1000 m) of the study area extends from $-0.008$ s$^{-1}$ to $0.008$ s$^{-1}$, with the majority between $-0.007$ s$^{-1}$ and $0.002$ s$^{-1}$. Several negative outliers ($<-0.01$ s$^{-1}$) become evident at very large depths (4–5 km).

The coefficient at the cubic nonlinear term of Eq. (1) has relatively small (but positive) values for waves of the first mode in the entire deep-water region of the South China Sea (Fig. 16b). This coefficient for waves of the second mode has also small absolute values in this area. It is positive only on the continental slope and turns negative in the entire eastern part of the sea. This coefficient for waves of the second mode has large positive values in selected locations of the Sulu Sea. The north-eastern shelf of the South China Sea is characterised by intermittent variations in the sign of this coefficient for both modes of internal waves. This area also shows the largest absolute values of this coefficient (0.001 m$^{-1}$s$^{-1}$) for both modes.



The histogram of different values of the coefficient at the cubic term of Eq. (2) is moderately skewed. It covers values from –0.001 m$^{-1}$s$^{-1}$ to 0.001 m$^{-1}$s$^{-1}$ and has a high and relatively narrow peak at zero values (Fig. 16c). The majority of the values of this coefficient are negative, in the range from –0.0005 m$^{-1}$s$^{-1}$ to zero. Positive values are scarce and small. For example, only 5 values are counted around 0.00025 m$^{-1}$s$^{-1}$ whereas some 75 negative values of the same magnitude exist at –0.00025

m$^{-1}$s$^{-1}$. Similarly to the coefficient at the quadratic term, the values of the coefficient at the cubic term of Eq. (2) are practically independent of the water depth for both modes (Fig. 17). Extensive scatter and the largest absolute values of this coefficient are characteristic of shallow areas. For water depths >1 km this coefficient is in the range from 0.00013 m$^{-1}$s$^{-1}$ to –0.0003 m$^{-1}$s$^{-1}$. As an exception, a few pixels in the Sulu Sea contain much larger positive (for waves of the second mode) or smaller negative (for waves of the first mode) values.

**3.3 Applicability of the asymptotic model for long internal waves in the South China Sea**

Gardner equation is, strictly speaking, only an asymptotically valid model for weakly nonlinear long internal waves. Thus, its applicability should be discussed for each particular environment and set of parameters of internal waves. The observed amplitudes of internal waves of the second mode in the shelf region of the South China Sea were in the range of 10–30 m. According to Yang et al. (2009, 2010), amplitudes of internal solitons of the second mode are about 20 m. We use the value

$A$ = 20 m for the evaluation of the applicability of the Gardner model (1) in the western part of the South China Sea located on the continental shelf together with the typical dimensional (physical) values of the coefficients of the quadratic and cubic nonlinear terms in Eq. (1). The above shows that in this region usually $\alpha$ = 0.01 s$^{-1}$ and thus the typical magnitude of the quadratic term is $\alpha A$ = 0.2 m s$^{-1}$. Similarly, the typical value of the coefficient at the cubic nonlinear term is $\alpha_1$ = 0.0005 m$^{-1}$s$^{-1}$ and the magnitude of this term is $\alpha_1 A^2$ = 0.2 m s$^{-1}$. The typical magnitude of the phase speed of linear long internal

waves of the second mode is $c \sim$ 0.4 m s$^{-1}$.

Therefore, both nonlinear terms of Gardner equation have an equal magnitude that is about half of the long wave speed of the internal waves of the second mode. This match signals that Gardner equation is clearly suitable for the description and analysis of properties, propagation and dynamics of internal waves of the second mode with amplitudes up to 20 m in the shallow-water part of the South China Sea. Even though the dynamics of such solitary waves and solitons is strongly

nonlinear, possible errors in the estimates of their parameters (first of all amplitude and phase speed) based on the Gardner model do not exceed 20% (Maderich et al, 2009, 2010). This conjecture is consistent with the practice of the use of such asymptotic models. This level of deviations of the estimates from the true values is commonly acceptable (Liu et al., 2004; Talipova and Pelinovsky, 2013; Talipova et al., 2014, 2015). As errors of this kind rapidly increase with increasing wave amplitude, the use of such models for smaller-amplitude waves is associated with much lower levels of errors.



# 4 Discussion and conclusions

The derived maps of various parameters of the governing quantities of the underlying model (such as the location of the maxima of the modal function) and the parameters of the weakly nonlinear models provide a new insight into qualitative features of the propagation and transformations of internal waves of the second mode in the South China Sea. The presented

climatologically valid distributions of the phase speed and coefficients at the nonlinear terms of Gardner equation (1) (or other equations of the family of KdV-type equations) may be used for express estimates of various parameters of internal waves of this kind. This includes *inter alia* evaluation of hydrodynamic loads on the seabed (and on offshore engineering structures) created by the propagation of such waves, forecasting of areas and depths strongly affected by the internal wave activity after intense wave generation events, and identification of regions with a very high probability that such waves will

break.

A promising development is the possibility of evaluation of the limiting amplitude of internal solitons that correspond to negative values of the coefficient at the cubic nonlinear term (Kurkina et al, 2011, 2017) as well as the amplitude of algebraic solitons that correspond to the positive values of this coefficient.

The main conclusions of the study are:

• Spatial distributions of all kinematic parameters of internal waves in the South China Sea (except for the coefficient at the cubic nonlinear term of Gardner equation) are qualitative similar for waves of both modes.

• The dispersive term of Gardner equation for internal waves of the second mode is about 3–4 times smaller than ~~this term~~ for waves of the first mode.

• The phase speed for internal waves of the second mode is about half of that for waves of the first mode.

• The coefficients at the quadratic and cubic terms of Gardner equation for internal waves of the second mode are practically independent of water depth.

• In contrast to internal waves of the first mode, the quadratic term of Gardner equation is mostly negative for waves of the second mode in the South China Sea.

*Data availability*. The data used in this study were extracted from the GDEM database.

*Competing interests.* The authors declare that they have no conflict of interest.

*Acknowledgements.* This study was initiated in the framework of the state task programme in the sphere of scientific activity

of the Ministry of Education and Science of the Russian Federation (projects No. 5.4568.2017/6.7 and No. 5.1246.2017/4.6) and financially supported by this programme, grants of the President of the Russian Federation (NSh-6637.2016.5 and MK-5208.2016.5), Russian Foundation for Basic Research (grant No. 16-05-00049), and institutional support IUT33-3 from the Estonian Research Council.



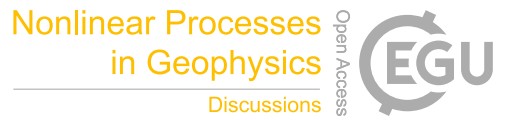

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

.





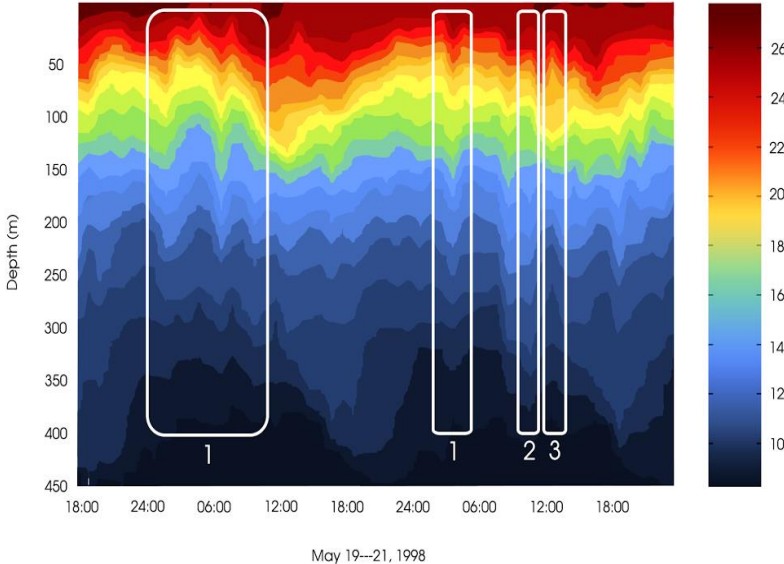

**Figure 1: Internal waves representing the first, second and third mode (Guo et al., 2006). Numbers at the boxes in the panel indicate the number of the relevant mode.**

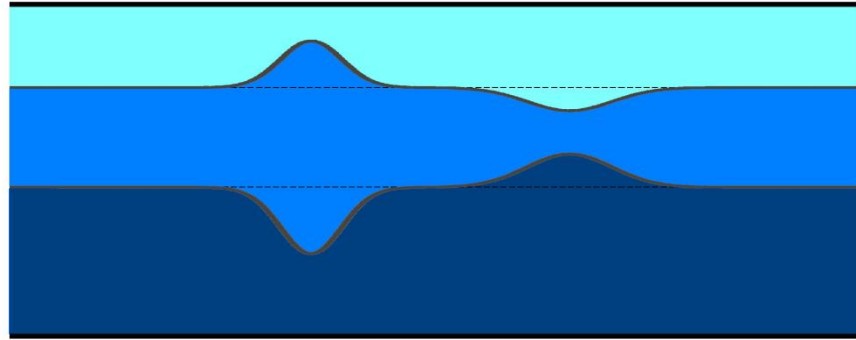

5 **Figure 2: Scheme of internal solitary waves of the second mode with positive (left) and negative (right) polarity.**



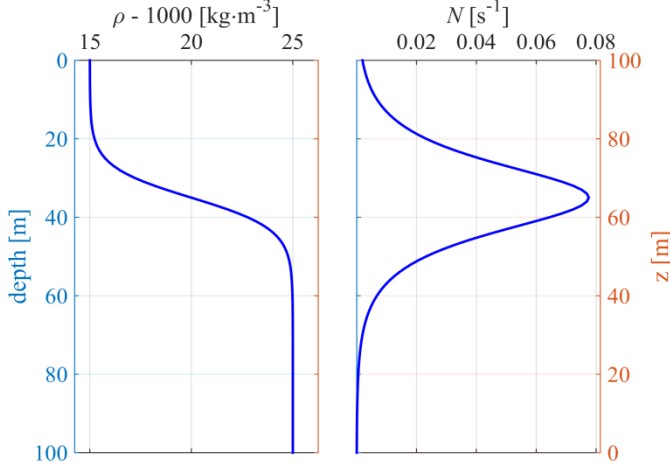

**Figure 3: An example of the vertical profile of the density (left) and Väisälä frequency (right) of a quasi-two-layer marine environment.**

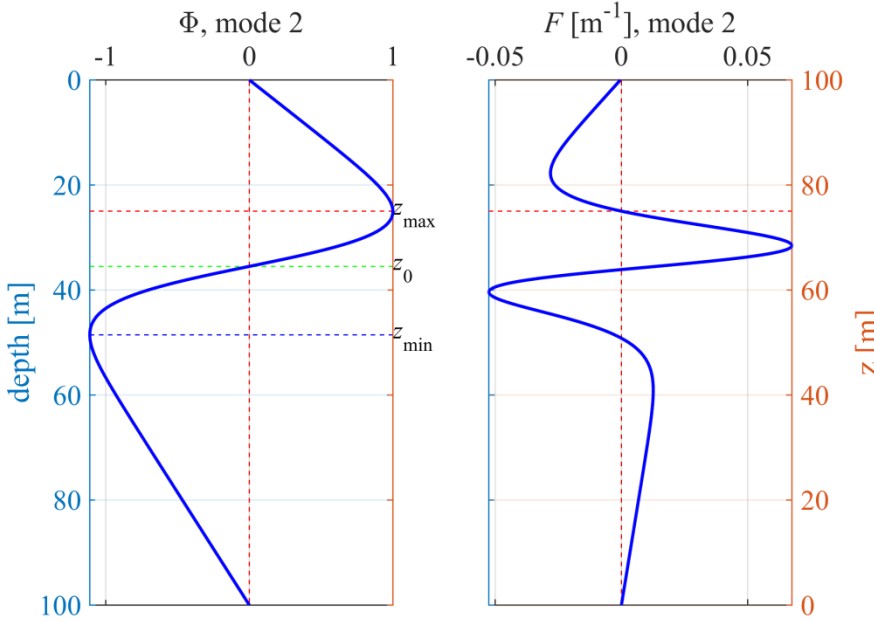

5   **Figure 4: The mode function and its nonlinear correction for internal solitary waves of the second mode with positive (left) and negative (right) polarity.**



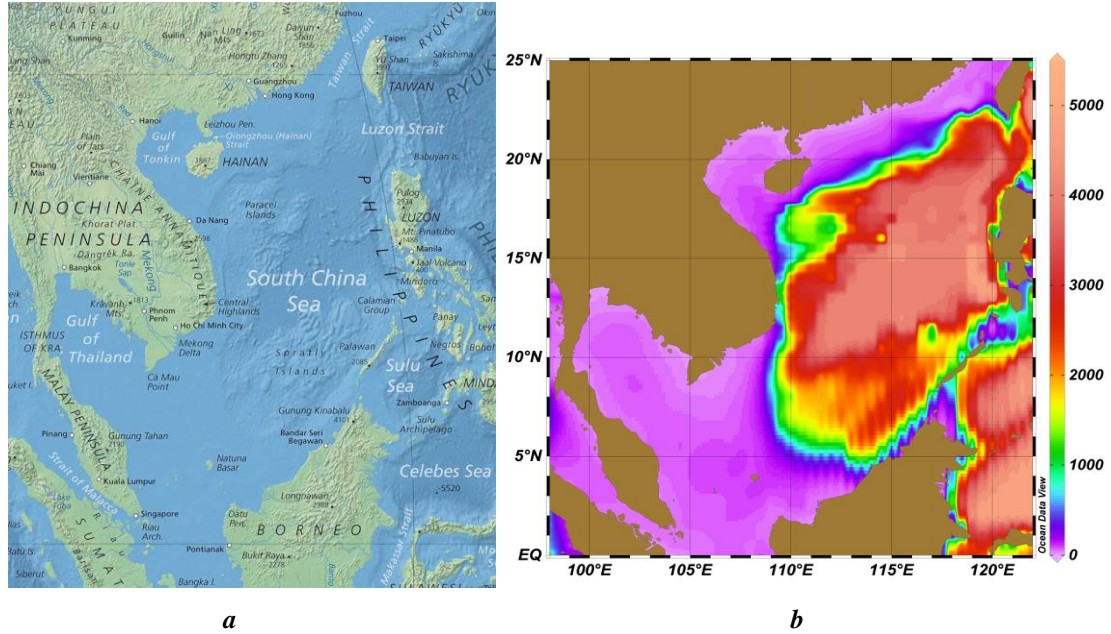

**Figure 5:** *a*) Location scheme of the South China Sea (http://www.nationsonline.org/oneworld/map/South-China-Sea-political-map.htm), *b*) bathymetry of the South China Sea extracted from the GDEM database (Carnes, 2009).

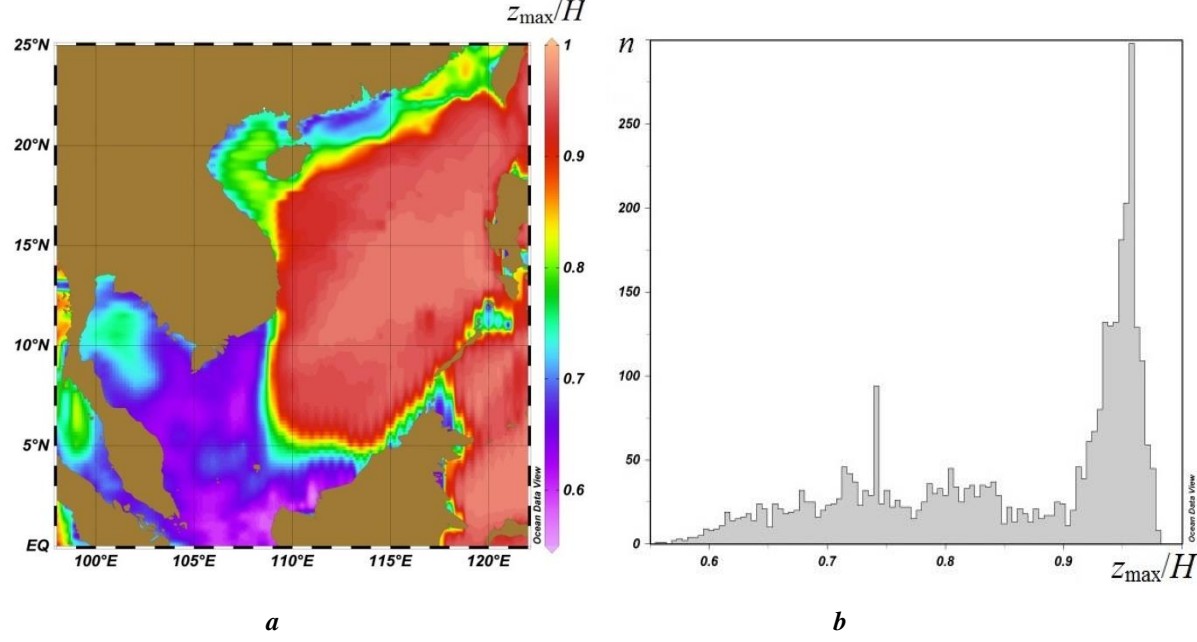

**Figure 6: a)** Map of the vertical location of the normalised depth $z_{max}/H$ of the maximum of the mode function in July in the South China Sea; b) histogram of values of $z_{max}/H$ in steps of 0.01 (*n* is the amount of values of $z_{max}/H$ in the relevant interval). The total number of pixels in the map is 3446.

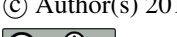


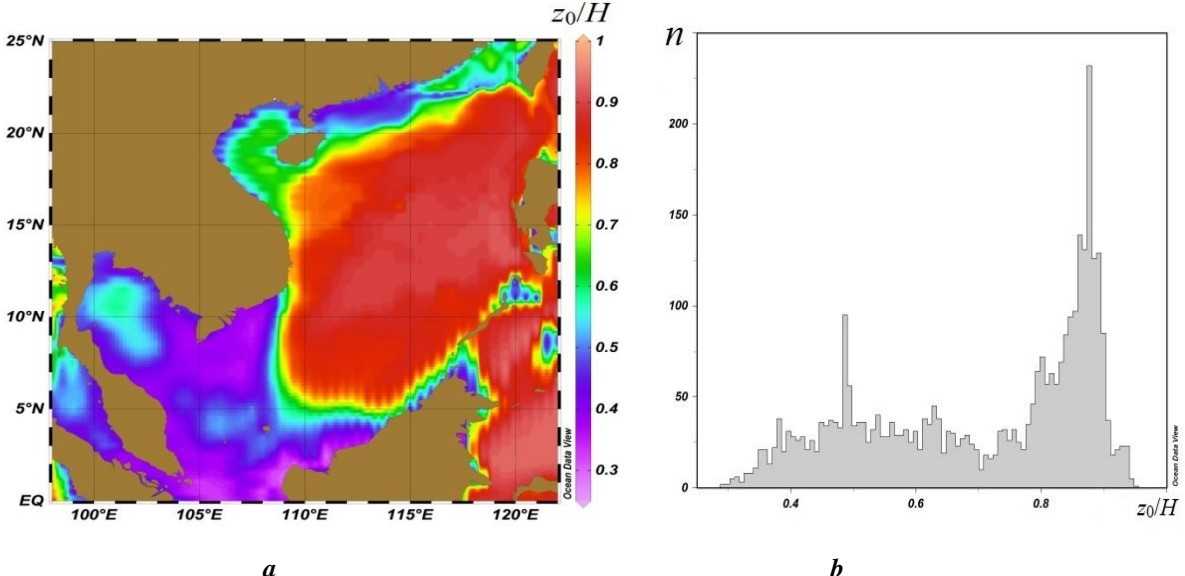

**Figure 7: a)** Map of the vertical location of the normalised zero-crossing depth $z_0/H$ of the mode function in July in the South China Sea; **b)** histogram of values of $z_0/H$. Notations are the same as for Fig. 6.

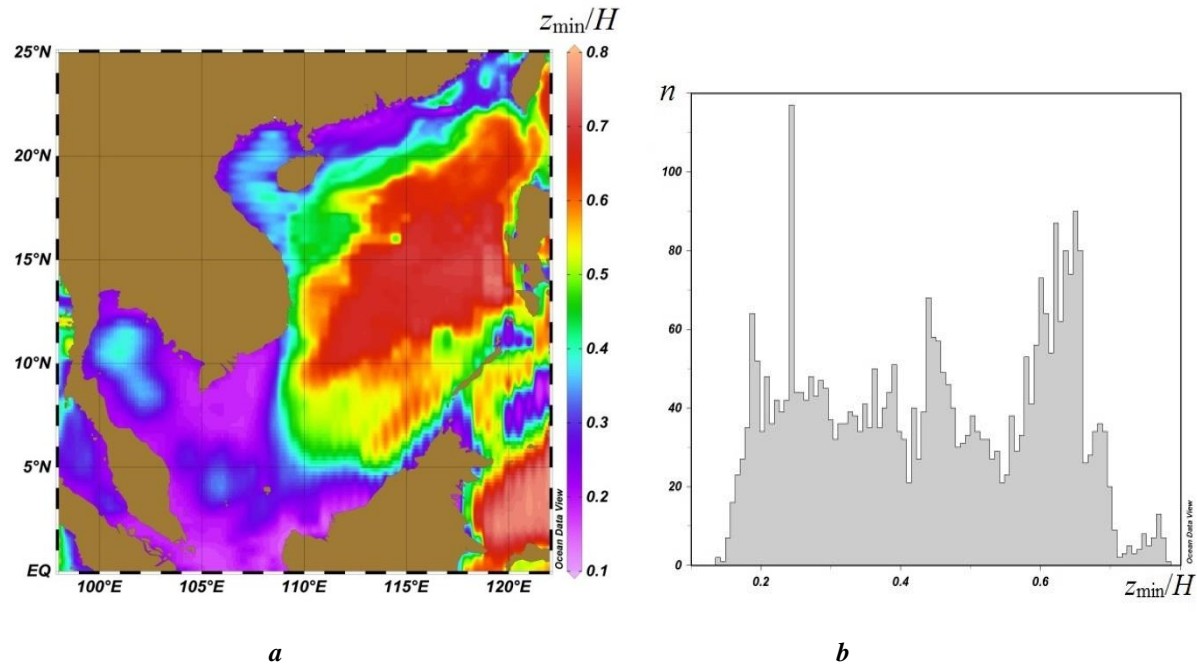

**Figure 8: a)** Map of the vertical location of the normalised depth $z_{min}/H$ of the minimum of the mode function in July in the South China Sea; **b)** histogram of values of $z_{min}/H$. Notations are the same as for Fig. 6.





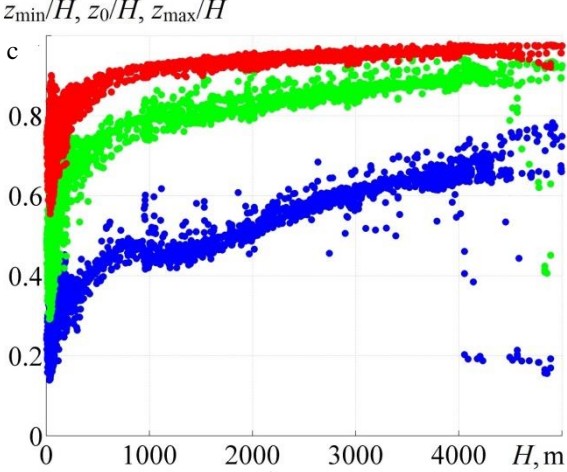

**Figure 9: Scatter-plot of normalised values of $z_{max}/H$ (red), $z_0/H$ (green), and $z_{min}/H$ (blue) against the total water depth in the South China Sea.**

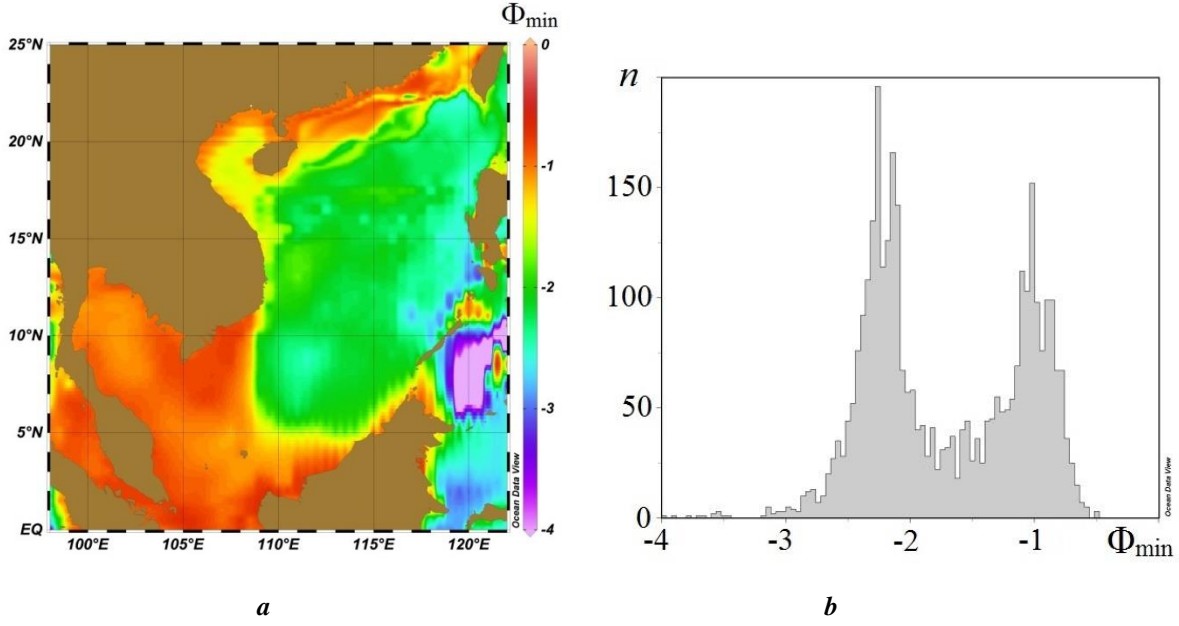

5                            *a*                                         *b*

**Figure 10: a) Map of normalised values $\Phi_{min}$ of the minimum of the mode function in July in the South China Sea; b) histogram of values of $\Phi_{min}$. Notations are the same as for Fig. 6.**





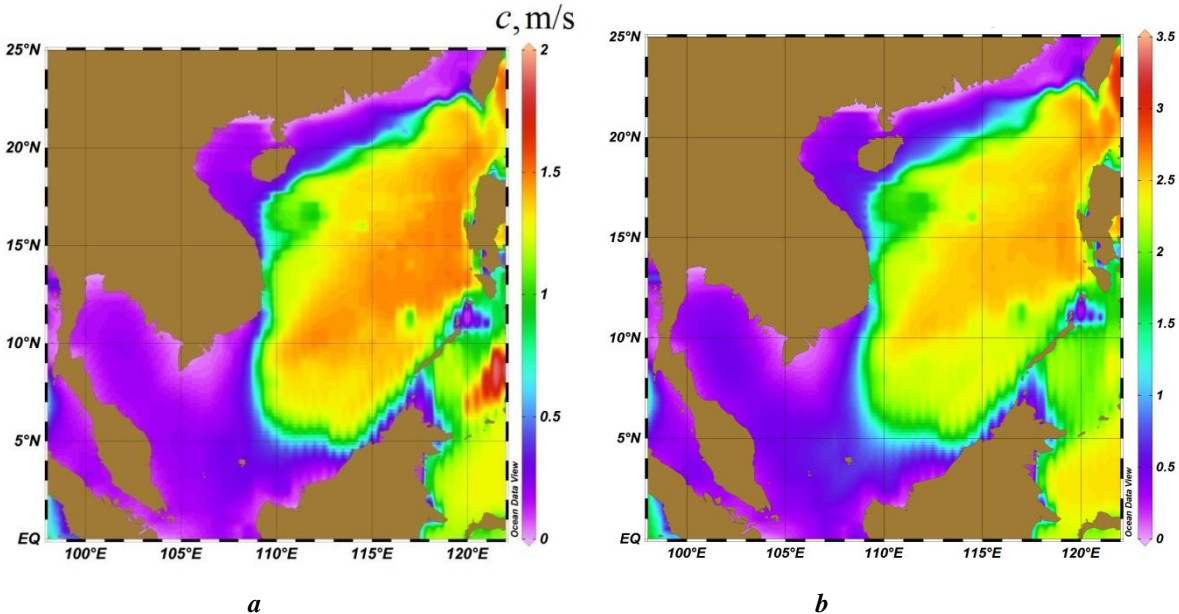

**Figure 11: Map of phase speeds of long linear internal waves of the second (a) and first (b) mode in July in the South China Sea.**

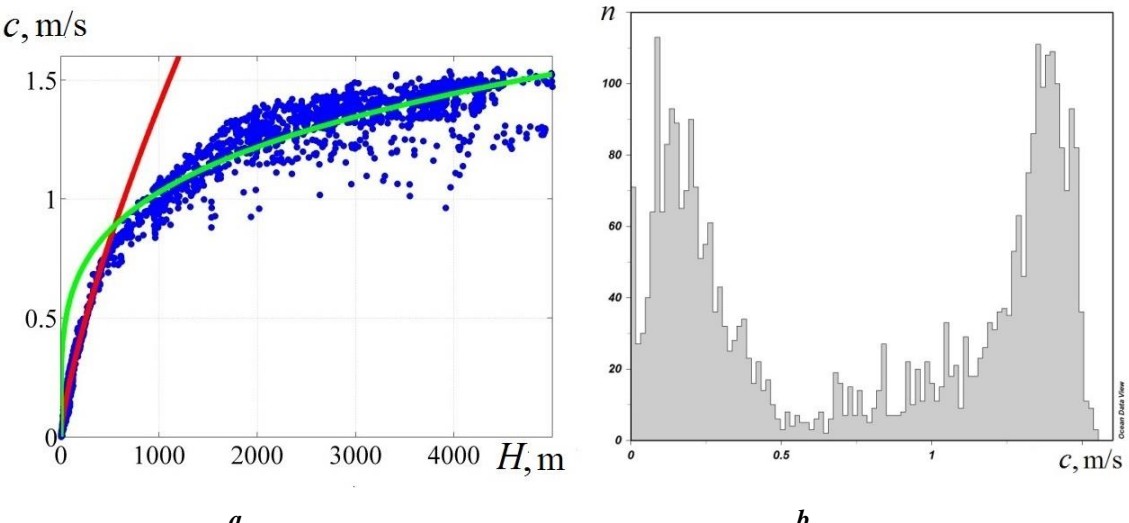

**Figure 12: a) Scatter-plot of phase speeds of linear long internal waves of the second mode for different water depths in July in the South China Sea. Red curve: approximation of the relationship between the phase speed and water depth for depths below 500 m with a power function (10); green curve: the same approximation for depths >500 m; b) histogram of different values of the phase speed of linear long internal waves of the second mode. Other notations are the same as for Fig. 6.**





*a*  *b*

*c*

5   **Figure 13: Map of coefficients at the dispersive term of Gardner equation for internal waves of the second mode (a) and first mode (b) in July in the South China Sea; c) scatter plot of this coefficient for internal waves of the second mode against water depth.**





**Figure 14: Map of coefficients at the quadratic term of Gardner equation for internal waves of the second mode (a) and first mode (b) in July in the South China Sea; c) histogram of this coefficient for internal waves of the second mode. Other notations are the same as for Fig. 6.**





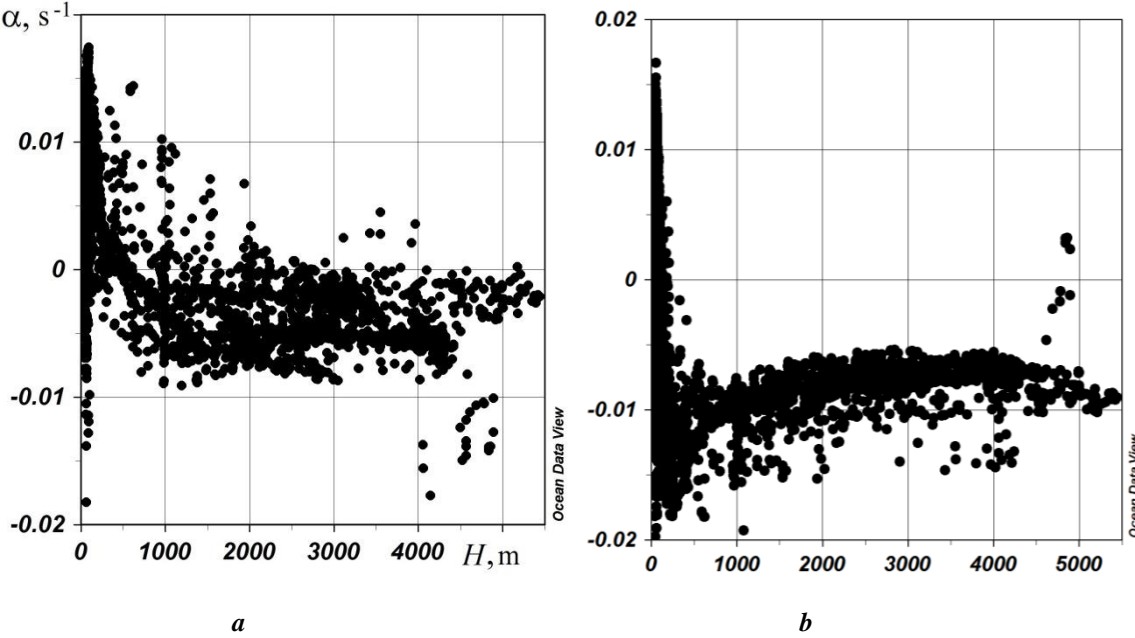

**Figure 15: Scatter diagram of coefficients at the quadratic term of Gardner equation against water depth for internal waves of the second mode (a) and the first mode (b).**



*a*

*b*

*c*

**Figure 16: Map of coefficients at the cubic term of Gardner equation for internal waves of the second mode (a) and first mode (b) in July in the South China Sea; c) – histogram of this coefficient for internal waves of the second mode. Other notations are the same as for Fig. 6.**




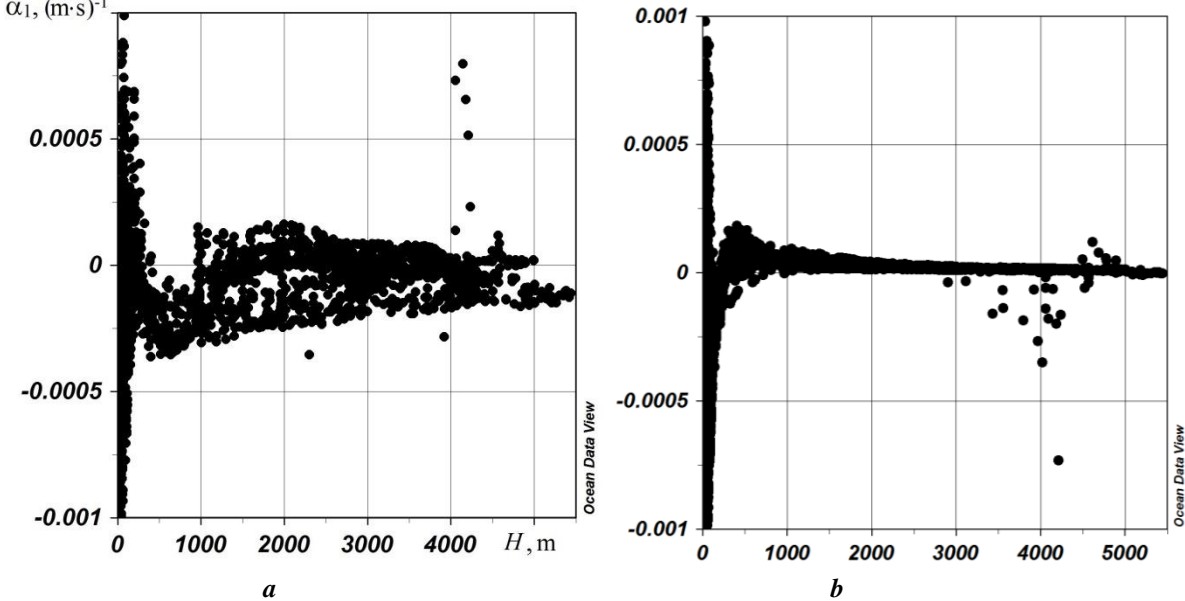

**Figure 17: Scatter diagram of coefficients at the cubic term of Gardner equation against water depth for internal waves of the second mode (a) and the first mode (b).**