# Peer review of "Kinematic parameters of internal waves of the second mode in the South China Sea"

_Nonlinear Processes in Geophysics, 2017_

## Referee Comment (RC1) · Anonymous Referee #1 · 17 Jun 2017

General comments:

Internal waves in the South China Sea (SCS) are among the largest that have been observed in the world's oceans and have been under intensive studies in the last two decades. The focus has been mostly on the first mode internal tides and internal solitary waves, whereas internal waves of the second mode have also been occasionally reported, and it seems that the latter are not unusual to be observed in the SCS. This paper summarises the spatial distributions of some kinematic parameters of second-mode internal waves in the SCS. The classice Gardner equation was employed in the study. Similar maps for the first mode internal waves have been constructed before, therefore, the current work can fill in the gap for the similar maps of the second mode internal waves. The maps can serve as a useful reference for quick assessment of

wave properties of the second mode. The work also fits the scientific scope of the journal. I would recomment publication of the work subject to some (minor) revision in order to meet the level of publication. I have my comments listed below.

Specific comments:

1) The authors examined the summer July conditions in the SCS, and I wonder if the authors have also looked into winter conditions? If the authors have done so, it would be nice if they could include a short paragraph illustrating the main differences or similarities. No worries if there is too much work. 2) P.1 L.24: Internal waves in the SCS are not excited by interactions of barotropic tides with the Kuroshio. Rather, they are generated by tide-topography interaction in the Luzon Strait, and Kuroshio can serve as a background current and modify the generating conditions. 3) P.1 L.10: is 'release of storm surges' a type of 'strong atmospheric disturbances'? 4) P.3 L.34: while I admit that it is meaningful to construct such 2D spatial kinematic maps of internal waves in the SCS as the authers have done, I don't think it can be as 'urgently needed'. 5) P.4 L22 and elsewhere: Brunt–Väisälä frequency. 6) P.9 L6-12: I don't think it correct and necessary to have such a discussion here about the radiation from the sun; suggest remove it. 7) P.12 L.25: 'Data availability' is not about the data that you used during your research, but it is about your own output data in the work; see the introduction here: http://www.nonlinear-processes-in-geophysics.net/about/data_policy.html 8) Figs. 6-8: please consider using the same colorbar scale in these three figures such that the readers can have a more direct comparion of the three different depths. This also applies to Figs. 11 & 13. 9) Figs. 14&16: please consider using a white-centered colorbar for a better visulisation of positive and negative values.

Technical corrections:

1) P.2 L.3: text correction. 2) P.2 L.5-6: sloppy; please reword. 3) P.3 L10: Carnes, 2009 4) P.6 L.20: quadratic term of Eq. (1)? 5) P.7 L.19: Kurkina et al. (2017) 6) P.8 L. 7: text correction. 7) P.8 L. 12: text correction. 8) P.9 L.24&25: Fig. 13C 9) P.11

L.24: ... solitions are strongly... 10) P.12 L.16: ...are qualitatively similar... 11) Fig. 12 caption: with a power function (8)

---

## Referee Comment (RC2) · Anonymous Referee #2 · 28 Jun 2017

On the manuscript "Kinematic parameters of internal waves of the second mode in the South China Sea" by O. Kurkina et al. This is one more work in the long series of publications of the group including the authors regarding different variants of the Gardner equation (GE) in application to the oceanic internal waves. It is devoted to calculation and mapping of GE parameters for South China sea for which, indeed, many observations of nonlinear (often strongly nonlinear) internal waves (IWs) have been published. The authors concentrate on the second IW mode which was observed albeit much less regularly than the first mode. It contains numerous details and maps of the second-mode wave structure the GE coefficients. The paper has a heuristic and perhaps some practical interest. It is written in a good English and potentially can be published. However, some serious questions should be addressed first. 1. The authors briefly discuss

applicability of the weakly nonlinear model. But it is not very convincing. If nonlinear corrections to phase velocity reach 50%, it may change the result significantly. I do not understand the 20% estimate for both amplitudes and velocities of solitons: usually one of them is taken empirically and the second is calculated based on that. Also how nonlinear corrections to the mode stricture affect the GE parameters? 2. Even more important is that after all the calculations no single comparison with the observed internal waves is given. What kind of nonlinear waves was observed at the second mode? Is there anything more specific in the literature than just box 2 in Fig. 1? If so it would add a justification for the detailed mapping, and vice versa. 3. For the deep parts of the SCS: what should be the wavelength of the IW to fit the long-wave approximation? Are the observed waves long enough for that? 4. Sorry if I missed that but what stratifications were used for calculation and mapping? It is mentioned about two pycnoclines but only a simple model is shown in Figure 3 (so only the total depth was varied according to the bathymetry?) 5. If alpha1 is mostly negative, and alpha2 is also negative, what kind of solitons can exist in this area? There is no flat-top solitons in this case. 6. The paper text is short but it is overloaded by figures. In my opinion, figures like 6-10 which show intermediate values are not necessary.

Minor notes: "solitary wave that interact elastically" -so it is a priori weak, integrable nonlinearity in a 100 m wave? How do you know? "highly energetic [?] internal waves of higher modes" -What do you mean by that? "The coefficients at the quadratic and cubic terms of Gardner equation for internal waves of the second mode are practically independent of water depth."- How can? Probably you mean the case of a deep lower layer? "are qualitative similar" -better "qualitatively" (this is about the only typo I noticed). Why "Vaisala" rather than "Brunt-Vaisala?"

---

## Referee Comment (RC3) · Anonymous Referee #3 · 6 Jul 2017

General comments:

The paper is devoted to the calculation and analysis of the wave speed of long internal waves of the second mode, and the respective coefficients of the weakly nonlinear model (Gardner equation) for the conditions of the South China Sea. The calculations use the GDEM database. As a result of this study, the authors have described important key trends in the behaviour of these parameters, which they have summarised at the end of section 4. A useful comparison has been made with the relevant coefficients for the waves of the first mode, and the authors have included a helpful discussion on their choice of the representative depth. Overall, I find that this is a useful study. However, I have some queries, and I hope that the authors will be able to address these queries in the revised version of the paper.

[Figure]

Specific comments:

1. In the Introduction it is stated that "These powerful disturbances are usually excited by interactions of barotropic tidal waves with the Kuroshio Current..." (Minor comment: 'excited' is probably better replaced with 'affected'). This statement seems to be in a direct contradiction with "The field of large-scale currents was ignored" on p. 7. Indeed, the modal equations on p. 4 and 5, and the coefficients of the Gardner equation on p.5 are calculated under the assumption that there is no background shear flow. However, the presence of the flow will change the very parameters calculated and analysed in the paper. Thus, the authors are asked to justify ignoring the currents.

2. In the discussion of the applicability of the Gardner equation for long internal waves in the South China Sea (section 3.3) the authors provide estimates for the terms in the bracket of the Gardner equation (1). This discussion seems to be incomplete. It would be useful to add estimates (or at least a discussion) for (a) the nonlinear and dispersive terms in (1), (b) the fifth and nonlinear dispersive terms which appear in the derivation of the higher-order KdV equation, but are neglected in this study. The authors are asked to clarify these points.

3. On p. 3 it is stated that "This feature makes it possible to use these models to isolate and identify principally new features of the dynamics of internal waves even if some details of the system are not reproduced..." The authors are asked to expand this discussion and briefly describe the main advantages and disadvantages of using the weakly nonlinear models of this type, rather than just referring to the literature.

4. On p. 10 it is stated that "Gardner equation is not applicable in locations where the coefficients at the quadratic term vanishes and one has to employ a modified KdV equation..." This is not clear to me. Gardner equation becomes the mKdV in this case, so, what is meant here?

Technical corrections:
1. A footnote with the web link to GDEM database would be useful to readers.

2. p. 1, "...solitons (solitary waves that interact elastically)" The comment in the bracket is not relevant in the context of this study, remove.

3. p. 5, "... are invariant with respect to the particular choice of z*..." is better replaced with "... do not depend on the particular choice of z*..."

4. Figure 4, caption is unclear. Please, check.

5. The list of references is too long for the size of the paper.

---

## Author Comment (AC3) · 2 Sep 2017

The comment was uploaded in the form of a supplement:
https://www.nonlin-processes-geophys-discuss.net/npg-2017-20/npg-2017-20-AC3-supplement.pdf

---

## Editor Comment (EC1) · V. I. Vlasenko (Editor) · 3 Sep 2017

Dear authors,

I've got now three reports on your paper and your responses to the reviewers' comments. Based on my personal reading and the documents provided I can conclude that the vast majority of the reviewers' remarks has been address adequately, so I'm pleased to encourage you submission of a revised manuscript corrected in line of the reviewers' recommendations.

I hope to hear from you soon,

Kind regards,

[Figure]

Handling Editor

---

## Author Comment (AC4) · 4 Sep 2017

Dear Prof. Vlasenko,

we have already uploaded the file with the text of the manuscript revised according to reviewers' recommendations.

Best regards, Authors

---

## Author Response (AR1)

We very much appreciate the overall positive attitude of all referees to our manuscript and thank them for particularly useful comments. The comments, questions and suggestions of the referees are presented in italics.

**Referee #1**

1. *"The authors examined the summer July conditions in the SCS, and I wonder if the authors have also looked into winter conditions?"*
During preparation of the manuscript we constructed maps of the coefficients and parameters of Gardner's equation for both January and July. These maps are mostly qualitatively and quantitatively similar to each other (except for the parameters $\alpha$ and $\alpha_1$).

[Figure]

[Figure]

[Figure]

Thus, seasonal changes in these parameters are generally fairly small and/or do not have a clear pattern. As the manuscript is already quite long and contains many figures (as stressed also by Referee #2), we decided not to include the maps for winter conditions into the paper. However, we added short comments on seasonality of these parameters into the manuscript (on pages 11 and 20).

"Even though several properties of water masses of the South China Sea exhibit extensive seasonal variations, this feature not necessarily becomes evident in terms of kinematic parameters of internal waves of the second mode. The maps of quantities that express the normalised stratification conditions ($z_{max}/H$, $z_0/H$, $z_{min}/H$, $\Phi_{min}$) and the linear parameters $c$ and $\beta$ for January (not shown) qualitatively almost coincide with similar maps for July. The match is almost perfect in the deeper area of the basin. The largest quantitative differences (on the order of 20%) occur in shallow areas of this sea. However, both coefficients at the nonlinear terms of Gardner equation have substantial seasonal variations. The values of the coefficient $\alpha$ at the quadratic term change insignificantly from July to January in deeper areas but are instead of very small values in July quite large (around and above 0.01 s$^{-1}$) in shallow areas. The values of the coefficient $\alpha_1$ at the cubic nonlinear term vary in a complicated manner between January and July"

2. " Internal waves in the SCS are not excited by interactions of barotropic tides with the Kuroshio."
Thank you for highlighting this issue. The wording is changed as follows: (page 1, 25)
"These powerful disturbances are usually excited by tide-topography interaction in the Luzon Strait where Kuroshio serves as a background current that may greatly modify the generating conditions. The resulting internal waves are further modified by numerous islands, seamounts and other bathymetric features in the Luzon Strait (Liu et al., 1998, 2004, 2006; Cai et al., 2002; Rump et al., 2004, 2015)."

3. *P.1 L.10: is 'release of storm surges' a type of 'strong atmospheric disturbances'?*
We mean that internal waves may be generated by storm surges also. We changed the sentence a little bit in order to make sure that we have in mind "indirect and/or delayed impact of such [strong atmospheric] disturbances (e.g., release of storm surges)".

4. P.*3 L.34: while I admit that it is meaningful to construct such 2D spatial kinematic maps of internal waves in the SCS as the authors have done, I don't think it can be as 'urgently needed'.*
We agree this adjective is too insistent. Thus, we deleted it and say now that these maps are "useful".

5. P.4 L22 *and elsewhere: Brunt–Väisälä frequency*
Thank you, this notion is correct.

6. P.9 L6-12: *I don't think it correct and necessary to have such a discussion here about the radiation from the sun; suggest remove it.*
We agree that this discussion is not really necessary in this paper, so omitted it and only mention in the revised version that spatial variations of the incoming radiation may have a certain impact on patterns of kinematic parameters of internal waves:
"Even though water depth is one of the most important factors governing the propagation speed of internal waves, stratification of water masses equally contributes to the properties of the propagation of internal waves. Its impact is apparently complemented by variations in the amount of incoming radiation from the Sun. These variations may be one of the reasons of the presence of the meridional pattern of the phase speed of internal waves of the second mode. This meridional pattern is well known for internal waves of the first mode (Talipova and Polukhin, 2001)."

7. P.12 L.25: *'Data availability' is not about the data that you used during your research, but it is about your own output data in the work*
It is changed as follows:
"The derived maps of the parameters as well as underlying data for histograms and scatter plots may be obtained from the authors in digital form via requests by e-mail"

*8. Figs. 6-8: please consider using the same colorbar scale in these three figures such that the readers can have a more direct comparison of the three different depths. This also applies to Figs. 11 & 13.*

Figs. 6–8. Thank you; we have redrawn the images as recommended by the Referee. The scales have the same range (0,1). Note that they became less informative as fewer colors are used for each quantity due to the narrower range.

[Figure]

[Figure]

Figs. 11 & 13. The colorbar range is decreased for mode I (right column) to match the scale with the one used for mode II (left column).

9. *Figs. 14 and 16: please consider using a white-centered colorbar for a better visualisation of positive and negative values.*

Thank you for this suggestion. However, it was not easy for follow. See an example for mode I below, left column. It turned out that white-centered blue-red palette was less effective (having less colours) to present the necessary information. Such white-centered plots are given for comparison, see below, right column. Thus, we chose another way. Namely, for a better visualization of negative and positive values we added the zero contours to the maps of nonlinear parameters (Figs. 14 and 16).

[Figure]

**Technical corrections**
*1) P.2 L.3: text correction:* unnecessary word removed *2) P.2 L.5-6: sloppy; please reword:* reworded as: It is likely that higher modes of long internal waves are often generated in the World Ocean. *3) P.3 L10: Carnes, 2009:* Of course, thank you *4) P.6 L.20: quadratic term of Eq. (1)?* Should be Eq. (1); thank you *5) P.7 L.19: Kurkina et al. (2017):* Corrected *6) P.8L. 7: text correction:* unnecessary letters removed *7) P.8 L. 12: text correction.* unnecessary word removed *8) P.9 L.24&25: Fig. 13C:* Thank you; corrected *9) P.11 L.24: ... solitons are strongly..:* reworded as

*"such solitary waves are strongly nonlinear".* *10) P.12 L.16: ...are qualitatively similar...:* Reworded to make the claim unambiguous *11) Fig. 12 caption: with a power function (8):* Thank you; corrected.

**Referee #2**

1. "*The authors briefly discuss applicability of the weakly nonlinear model. But it is not very convincing. If nonlinear corrections to phase velocity reach 50%, it may change the result significantly.*
Thank you; we just forgot to describe the estimate of the upper limit of wave amplitudes for which the Gardner equation is suitable in the South China Sea conditions. Thus, we added: "In the light of estimates of Maderich et al (2009, 2010), the presented relationships signal that the Gardner equation is suitable for the description and analysis of properties, propagation and dynamics of internal waves with the amplitude of up to 20 m in the South China Sea conditions."

*I do not understand the 20% estimate for both amplitudes and velocities of solitons: usually of them is taken empirically and the second is calculated based on that.*
Sure, we deleted "velocities"

*Also how nonlinear corrections to the mode structure affect the GE parameters?*"
First of all, it concerns the coefficient at the cubic nonlinear term (Eq. 7). Without such a correction (e.g. within the two-layer model) this coefficient transforms into the expression obtained by Kakutani & Yamasaki, (1978)

$$\alpha_1 = -\frac{3c}{8h_1^2 h_2^2}(h_1^2 + h_2^2 + 6h_1 h_2)\text{, which is negative everywhere.}$$

For more complicate stratifications it may be positive, negative or vanish. Such variations are studied based on several examples of three-layer stratification in (Grimshaw et al, 1997; Kurkina et al., 2015). We feel that it is not really necessary to repeat their arguments and thus decided to add the following remark (P.5, L.20)
"The impact of stratification and mode correction on the value and sign of the coefficient at the cubic term in Gardner equation is analysed in detail in (Grimshaw et al, 1997; Kurkina et al., 2015)."
We also corrected a typo in Eq. (7).

2. "*Even more important is that after all the calculations no single comparison with the observed internal waves is given. What kind of nonlinear waves was observed at the second mode? Is there anything more specific in the literature than just box 2 in Fig. 1? If so it would add a justification for the detailed mapping, and vice versa.*"
We agree that observations of internal solitary waves of the second mode are scarce in the South China Sea; however, this is the general feature of many seas and oceans. However, we do not fully agree with the critics of Referee #2 as we have discussed examples of solitary waves of concave and convex type that have been observed in the Western South China Sea (Yang et al., 2009, 2010).
As there are, to our knowledge, no more descriptions of observations of this type in the international scientific literature, we are not able to expand this part of the manuscript.
The only one comparison what we can do is to visualize Gardner's solitons with different amplitudes and coefficients that are calculated maximally close to the point of the above-mentioned observation. After that we may try to compare the shape of the Gardner soliton of the corresponding amplitude with the shape of the observed solitary waves. Such a comparison makes sense if there are simultaneous recordings of temperature and salinity (or density) profiles in the measurement area. The aim of our work was to establish certain average seasonal characteristics of the internal waves of

the second mode and a general analysis of their possible combinations that determine the type, shape, and velocity of localized waves.

3. "*For the deep parts of the SCS: what should be the wavelength of the IW to fit the long-wave approximation? Are the observed waves long enough for that?* "

An internal wave can actually be considered as a long wave when the relation $\beta k^2 \ll c$, which implicitly depends on the sea water density stratification and includes dispersive coefficient $\beta$ and long linear internal wave phase speed $c$, is satisfied. Here $k = 2\pi/\lambda$ is the characteristic wavenumber and $\lambda$ is the characteristic length scale. (In the opposite case the waves are of intermediate length or short waves).

This condition can be much more gentle than just $H \ll \lambda$. For example, solitary waves of the second mode were observed in the shelf region of the South China Sea with the depth of about 300 m (Yang et al., 2009, 2010). The dispersion coefficient there is about 1500 m$^3$/s and the speed of propagation is about 0.5 m/s (Fig.12a). Therefore, waves longer than 350 m can be already considered as long waves in this region. The length of internal solitary wave of the second mode detected in (Yang et al, 2009) is about 350 m and this wave, consequently, can be treated as a long wave.

In deeper parts of the South China Sea internal waves can propagate either as a practically linear (sinusoidal) disturbances or as a steepening baroclinic tide, predominantly of the first mode. There may also exist reflected localized disturbances of the second mode generated on the unevenness of the bottom from the waves of the first mode. When such waves propagate in an inhomogeneous medium, they will undergo slow adiabatic changes or faster transformations. When they propagate into the region of increasing depth, their amplitudes will decrease and their lengths will increase, so they still can be considered as long waves.

As this material is, in essence, classic, we do not feel necessary to reflect it in the manuscript.

4. "*Sorry if I missed that but what stratifications were used for calculation and mapping? It is mentioned about two pycnoclines but only a simple model is shown in Figure 3 (so only the total depth was varied according to the bathymetry?)*"

We used averaged stratifications from GDEM climatology, from which we numerically evaluate the local stratification. We make this clear in the revised version already in the abstract. Fig. 3 is just schematic one to show the types of polarity of second-mode internal waves.

5. "*If alpha is mostly negative, and alpha1 is also negative, what kind of solitons can exist in this area? There is no flat-top solitons in this case. 6. The paper text is short but it is overloaded by figures. In my opinion, figures like 6-10 which show intermediate values are not necessary.*"

Based on the weakly nonlinear theory (Gardner's equation), if $\alpha_1$ is negative, the solitary wave may change their amplitude from 0 to the limiting value $A_{lim} = \alpha/\alpha_1$. The polarity of such solitary wave depends on the sign of the quadratic nonlinear term expressed by $\alpha$. A solitary wave with an amplitude close to the limiting amplitude is has a shape of the "thick" or "table-top" solitary wave. If $\alpha < 0$, a table-top solitary wave of depression may exist (Pelinovsky et al., 2007).

As for figures, we are in a complicated position because Referee #1 wishes to see some additional images. We feel that Figures 6–10 are still important as they present some flavor of the physical structure of water masses in the study area and carry important information about the frequency of occurrence of the relevant normalized background conditions.

Minor notes: *"solitary wave that interact elastically" -so it is a priori weak, integrable nonlinearity in a 100 m wave?*
This remark is deleted.

*How do you know? "highly energetic [?] internal waves of higher modes" -What do you mean by that?*

We deleted the words "highly energetic".

*"The coefficients at the quadratic and cubic terms of Gardner equation for internal waves of the second mode are practically independent of water depth."- How can? Probably you mean the case of a deep lower layer? "are qualitative similar" -better "qualitatively" (this is about the only typo I noticed).*
The text is changed as follows: "The coefficients at the quadratic and cubic terms of Gardner's equation for internal waves of the second mode mainly depend on the stratification and much less on the total water depth." Also, the typo has been corrected.

*Why "Vaisala" rather than "Brunt-Vaisala?"*
Replaced to "Brunt-Väisäla" as also recommended by Referee #1.

**Referee #3**

1. *In the Introduction it is stated that "These powerful disturbances are usually excited by interactions of barotropic tidal waves with the Kuroshio Current..." (Minor comment: 'excited' is probably better replaced with 'affected'). This statement seems to be in a direct contradiction with "The field of large-scale currents was ignored" on p. 7. Indeed, the modal equations on p. 4 and 5, and the coefficients of the Gardner equation on p.5 are calculated under the assumption that there is no background shear flow. However, the presence of the flow will change the very parameters calculated and analysed in the paper. Thus, the authors are asked to justify ignoring the currents.*

Thank you for highlighting this issue. The wording is changed as follows: (page 1, 25)
"These powerful disturbances are usually excited by tide-topography interaction in the Luzon Strait where Kuroshio serves as a background current that may greatly modify the generating conditions. The resulting internal waves are further modified by numerous islands, seamounts and other bathymetric features in the Luzon Strait (Liu et al., 1998, 2004, 2006; Cai et al., 2002; Rump et al., 2004, 2015)."

*"These powerful disturbances are usually excited by tide-topography interaction in the Luzon Strait, and Kuroshio can serve as a background current. They are further modified by numerous islands, seamounts and other bathymetric features in the Luzon Strait (Liu et al., 1998, 2004, 2006; Cai et al., 2002; Rump et al., 2004, 2015)."*
We do agree that the Kuroshio Current may affect the wave generation, but it does not explicitly affect the coefficients of the model in the South China Sea. We have added the relevant comment into the text and commented this issue also at the end of the body text.

2. *In the discussion of the applicability of the Gardner equation for long internal waves in the South China Sea (section 3.3) the authors provide estimates for the terms in the bracket of the Gardner equation (1). This discussion seems to be incomplete. It would be useful to add estimates (or at least a discussion) for (a) the nonlinear and dispersive terms in (1), (b) the fifth and nonlinear dispersive terms which appear in the derivation of the higher-order KdV equation, but are neglected in this study. The authors are asked to clarify these points.*
Gardner's equation is derived under the assumption that the coefficient at its quadratic term may vanish and change sign. When this coefficient $\alpha = 0$, the cubic nonlinear term is the main nonlinear term and the equation, formally, transforms into the mKdV equation. When this coefficient tends to zero but does not vanish yet, one can write $\alpha = \varepsilon\chi$, where $\varepsilon$ is a small parameter. The transformation $X = \sqrt{\varepsilon}\, x\; T = \varepsilon\sqrt{\varepsilon}\, t$ converts then the second order asymptotic equation

$$\frac{\partial \eta}{\partial t} + \alpha \eta \frac{\partial \eta}{\partial x} + \beta \frac{\partial^3 \eta}{\partial x^3} + \varepsilon \left( \alpha_1 \eta^2 \frac{\partial \eta}{\partial x} + \gamma_1 \eta \frac{\partial^3 \eta}{\partial x^3} + \gamma_2 \frac{\partial \eta}{\partial x} \frac{\partial^2 \eta}{\partial x^2} + \beta_1 \frac{\partial^5 \eta}{\partial x^5} \right) = 0 \text{, where the cubic term is in the}$$

second order, into the equation

$$\frac{\partial \eta}{\partial T} + \left( \chi \eta + \alpha_1 \eta^2 \right) \frac{\partial \eta}{\partial X} + \beta \frac{\partial^3 \eta}{\partial X^3} + \varepsilon \left( \gamma_1 \eta \frac{\partial^3 \eta}{\partial X^3} + \gamma_2 \frac{\partial \eta}{\partial X} \frac{\partial^2 \eta}{\partial X^2} \right) + \varepsilon^2 \beta_1 \frac{\partial^5 \eta}{\partial X^5} = 0 \,.$$

The quadratic and cubic terms are in the same order in this framework and the other terms are in the next order.

As this procedure is widely used in the theory of weakly nonlinear waves, we do not feel necessary to comment it in the manuscript. A detailed discussion of this procedure and implications can be found in (Pelinovsky E.N., Slunyaev A.V., Polukhina O.E., Talipova T.G. Internal Solitary Waves. In: *Solitary Waves in Fluids* (ed. by R.Grimshaw), WIT Press, Southampton, Boston, 2007, 85–110). We included a reference to this source and tell now: "The case when both nonlinear terms are small is discussed in detail by Pelinovsky et al. (2007)."

*3. On p. 3 it is stated that "This feature makes it possible to use these models to isolate and identify principally new features of the dynamics of internal waves even if some details of the system are not reproduced..." The authors are asked to expand this discussion and briefly describe the main advantages and disadvantages of using the weakly nonlinear models of this type, rather than just referring to the literature.*

We add the following paragraph:
"For example, a new kind of quasi-steady nonlinear internal waves (so-called breathers) has been predicted using the framework of Gardner's equation. The possibility of generation of such phenomena by solitary waves of the second mode and the basic properties of its long-term propagation have been obtained in a numerical "wave tank" using Euler's equations (Lamb et al., 2007; Terletska et al., 2016). Several features of the process of generation of table-top solitary waves were also extracted based on Gardner's equation (Kurkina et al., 2016). The effect of a change in the polarity of solitary waves predicted by the asymptotic theory has been repeatedly observed in various areas including the South China Sea. It is however inevitable that many specific features and details (e.g. radiation of short waves, properties of strongly nonlinear disturbances or breaking of solitonic structures) cannot be reproduced using equations for weakly nonlinear waves and specific configurations of stratification may require the use of higher-order analysis and equations."

*4. On p. 10 it is stated that "Gardner equation is not applicable in locations where the coefficients at the quadratic term vanishes and one has to employ a modified KdV equation..." This is not clear to me. Gardner equation becomes the mKdV in this case, so, what is meant here?*
The text is changed as follows: "Gardner's equation transforms into the modified KdV equation in locations where the coefficient at the quadratic term vanishes and one has to use this equation in order to properly describe weakly nonlinear dynamics of internal waves in such regions."

Technical corrections

*1. A footnote with the web link to GDEM database would be useful to readers.*
We inserted it into the body text:
https://data.nodc.noaa.gov/cgi-bin/iso?id=gov.noaa.nodc:9600094

*2. p. 1, "...solitons (solitary waves that interact elastically)" The comment in the bracket*

*is not relevant in the context of this study, remove.*
It is deleted as recommended also by Referee #2

*3. p. 5, "... are invariant with respect to the particular choice of z*..." is better replaced with "... do not depend on the particular choice of z*..."*
Yes, this is easier to understand for many readers

*4. Figure 4, caption is unclear. Please, check.*
Thank you; we reformulated the caption. Also, we unified description of the axes and labels for the two panels of Fig. 4.

*5. The list of references is too long for the size of the paper.*
We are again in an intricate position as other referees implicitly wished to see even more references. Due to the revision we had to add some references. However, we deleted several sources that are not particularly critical for this manuscript: (Ramp et al., 2004; Talipova et al., 1998; Talipova and Pelinovsky, 2013).

[revised manuscript text omitted]